

# GAN-argcPredNet v2.0: A Radar Echo Extrapolation Model based on Spatiotemporal Process Intensification

Kun Zheng[1*], Qiya Tan[1*], Huihua Ruan[2], Jinbiao Zhang[2], Cong Luo[3], Siyu Tang[3], Yunlei Yi[4], Yugang Tian[1], Jianmei Cheng[5]

[1]School of Geography and Information Engineering, China University of Geosciences, Wuhan, 430074, China
[2]Guangdong Meteorological Observation Data Center, Guangzhou, 510080, China
[3]Guangdong Meteorological Observatory, Guangzhou, 510080, China
[4]Wuhan Zhaotu Technology Co. Ltd., Wuhan, 430074, China
[5]School of Environmental Studies, China University of Geosciences, Wuhan, 430074, China
*These authors have contributed equally to this work

*Correspondence to*: Kun Zheng (ZhengK@cug.edu.cn); Huihua Ruan (ruanhuihua@163.com)

**Abstract.** Precipitation nowcasting has important implications for urban operation and flood prevention. Radar echo extrapolation is the common method in precipitation nowcasting. Using deep learning models to extrapolate radar echo data has great potential. The increase of lead time leads to a weaker correlation between real rainfall evolution and generated images. The evolution information is easily lost during extrapolation, which is reflected as echoes attenuation. Existing models, including Generative Adversarial Network (GAN)-based models, are all difficult to reduce loss and curb attenuation, which results in insufficient rainfall prediction accuracy. Aim to the problem, a Spatiotemporal Process Intensification Network (GAN-argcPredNet v2.0) based on GAN-argcPredNet v1.0 is designed. GAN-argcPredNet v2.0 reduces the loss by intensifying the influence of the previously input evolution information. A Spatiotemporal Information Changes Prediction (STIC-Prediction) network is designed as generator. With the intensification of echo feature sequence, the generator focuses on the spatiotemporal variation and generates more accurate images. Furthermore, discriminator is a Channel-Spatial Convolution (CS-Convolution) network. The discriminator intensifies the discrimination of echoes information by strengthening spatial information of single image. Identification results are fed back to the generator, which reduces the loss of important evolutionary information. The experiments are based on the radar dataset of South China. The results show that GAN-argcPredNet v2.0 performs better than other models. In heavy rainfall prediction, compared with baseline, the Probability of Detection (POD), the Critical Success Index (CSI), and the Heidke Skill Score (HSS) increase by 24.8 %, 22.2 % and 21.5 % respectively. The False Alarm Ratio (FAR) decreases by 3.76 %.

## 1 Introduction

Accurate precipitation nowcasting, especially heavy precipitation nowcasting, plays a key role in hydrometeorological applications such as urban-operation safety and flash-flood warnings (Liu et al., 2015). It can effectively prevent the hazards and losses caused by heavy precipitation to economy and people (Luo et al., 2020).





Radar echo extrapolation is the method most often used to nowcast precipitation (Reyniers, 2008). The essence is tracking areas of reflectivity to derive motion vectors, and then using the motion vectors to determine future location of the reflectivity (Austin and Bellon, 1974). Traditional radar echo extrapolation methods include cross-correlation, individual radar echo-tracking and the optical flow method (Bowler et al., 2004). As the storm evolution like merging, splitting, growth and decay, traditional methods are difficult to predict accurately. Using deep learning models to extrapolate has great potential (Foresti et al., 2019). Deep learning has powerful nonlinear mapping ability. By studying the motion process from a large number of historical radar echo images, deep learning has better results (Shi et al., 2015; Pan et al., 2021). Compared with other deep learning models, the Generative Adversarial Network (GAN)-based models have great advantages in generating high quality echo images (Tian et al., 2020; Xie et al., 2022).

The radar echo images are predicted for a future period based on the real echo sequence. In the deep learning models, the increase of lead time leads to a weaker correlation between the real images at the front of sequence and the generated image. The influence of real echoes evolution is rapidly diminishing. In this process, the models lose rainfall evolution information. It is reflected as echoes attenuation on generated images. Due to the smaller percentage of heavy rainfall areas, the attenuation is more severe. Existing deep learning models, including GAN-based models, lack the method to curb attenuation, which leads to low accuracy in predicting heavy rainfall.

In this work, a Spatiotemporal Process Intensification Network (GAN-argcPredNet v2.0) is designed based on Generative Adversarial Advanced Reduced-Gate Convolutional Deep Predictive Coding Network (GAN-argcPredNet v1.0) (Zheng et al., 2022), which reduces loss and curbs attenuation. In GAN-argcPredNet v2.0, a Spatiotemporal Information Changes Prediction (STIC-Prediction) network is designed as the generator. The generator focuses on the spatiotemporal variation of radar echo feature sequence. The more accurate images are generated by intensifying the spatiotemporal evolution of previous inputs. Furthermore, discriminator is a Channel-Spatial Convolution (CS-Convolution) network. By focusing on radar echo features from spatial and channel dimensions, the discriminator enhances the ability to identify echoes information. In this way, the generator can better retain evolution information. Generator and discriminator are trained against each other to have accurate rainfall prediction. The experiments are based on the radar dataset of South China. The results show that GAN-argcPredNet v2.0 performs better than other models, e.g., Convolution Gated Recurrent Unit (ConvGRU), Convolutional Long Short-Term Memory (ConvLSTM), Generative Adversarial ConvGRU (GA-ConvGRU) and GAN-argcPredNet v1.0.

## 2 Related Work

Radar echo extrapolation based on deep learning has better performance than traditional methods. Sequence network and GAN are two common neural networks in radar echo extrapolation. Previous studies showed that GAN performed better in enhancing image quality. However, the prediction of heavy rainfall is still insufficient.





## 2.1 Radar-based precipitation nowcasting

Extrapolation based on radar echo images is a common method for precipitation nowcasting. Eulerian persistence is a naive
extrapolation method, which is based on using the latest available observation as a prediction. Eulerian persistence is quite a
powerful model for very short lead times (Ayzel et al., 2019). Traditional extrapolation techniques include cross-correlation
and individual radar echo-tracking (Pierce et al., 2004; Liguori and Rico-Ramirez, 2014). Thunderstorm Identification,
Tracking, Analysis, and Nowcasting (TITAN) is a classical centroid tracking algorithm (Dixon and Wiener, 1993). The
algorithm achieves precipitation nowcasting through real-time tracking and automatic identification of individual storm. The
tracking performance of TITAN is poor during multi-cell storms. Then, an enhanced TITAN (ETITAN) is proposed (Han et
al., 2009). By combining cross-correlation and individual radar echo-tracking, ETITAN achieves more accurate tracking and
prediction. Cross-correlation method, however, has significantly lower prediction accuracy when echoes change rapidly. The
optical flow method achieves local prediction by treating echoes motion as fluid (Sakaino, 2013). Pyramid Lucas-Kanade
Optical Flow method (PPLK) achieves better extrapolation by introducing a typical local differential optical flow method
(Liu et al., 2015). As radar echoes continuously evolve, the invariance assumption of the optical flow method cannot be
satisfied. The extrapolation accuracy is affected. Besides, these traditional methods fail to utilize the large amounts of
historical images.

## 2.2 Sequence networks for image sequence prediction

Deep learning has powerful nonlinear mapping ability and makes full use of historical data. Radar echo extrapolation can be
regarded as an image sequence prediction problem. Therefore, the problem can be solved by implementing an end-to-end
sequence learning method (Sutskever et al., 2014; Shi et al., 2015). ConvGRU learns video features through convolution
operation, which realizes sparse connection of model units (Ballas et al., 2015). Convolution operation is also used in
ConvLSTM. By replacing the steps of internal data state transformation in LSTM, ConvLSTM can better extract features
(Shi et al., 2015). Convolutional recursive structure is position invariant, which is not consistent with the natural variation
motion. Trajectory GRU (TraijGRU) is further proposed (Shi et al., 2017). Both LSTM and GRU models have long-term
memory. However, this capability is limited to historical spatial information and has a limited memory. RainNet builds a
convolutional network architecture in precipitation nowcasting, which avoids the brittleness of LSTM structure (Ayzel et al.,
2020). This new structure still fails to address the information loss. Meanwhile, the realistic details of images are also
insufficient in these models.

Attention mechanism is also often used in sequential networks. By learning the importance of different image parts, attention
mechanisms can improve prediction accuracy. For example, the self-attention mechanism combines the spatial relationships
of different locations and reinforces important areas (Wang et al., 2018a). Eidetic 3D LSTM (E3D-LSTM) introduces self-
attention to intensify long-term memory in LSTM (Wang et al., 2018b). However, it lacks attention in the channel dimension.
Interaction Dual Attention LSTM (IDA-LSTM) expands spatial and channel attention based on self-attention to better obtain

the representation (Luo et al., 2021). As the high hardware load, self-attention is hard to train high resolution images. Convolutional Block Attention Module (CBAM) was developing simultaneously as a less computational attention mechanism. It can be flexibly applied in sequential networks (Woo et al., 2018). These attention methods reinforce spatial information. In sequence prediction, the temporal information is also important, but these methods fail to reinforce it. For radar echo extrapolation, it reflects as a lack of intensification to rainfall evolution information.

## 2.3 GAN-based Radar Echo Extrapolation

At present, high quality extrapolation is mostly achieved by GAN (Tian et al., 2020; Xie et al., 2022). GAN consists of a generator and a discriminator, which has powerful data generation capabilities (Goodfellow et al., 2020). This is because the model with anti-loss can better realize multi-modal modeling (Lotter et al., 2016). For instance, Deep Generative Models of Rainfall (DGMR) generates more accurate reflectivity by adversarial training (Ravuri et al., 2021). GAN is also used to
generate realistic details for broader extrapolation range (Chen et al., 2019). GA-ConvGRU uses ConvGRU as the generator. By implementing multi-modal data modeling, the images quality is far better than ConvGRU (Tian et al., 2020). A number of studies contribute to improving the stability of GAN training. Energy-Based Generative Adversarial Forecaster (EBGAN-Forecaster) combines convolution structure and codec framework to improve stability (Xie et al., 2022). Also, our proposed GAN-argcPredNet v1.0 has more advantages in improving the predicted echoes details and stabilizing GAN training (Zheng
et al., 2022). As the lack of curbing echoes attenuation, all these models have limited accuracy in rainfall prediction, especially heavy rainfall prediction.

## 2.4 Summary

In precipitation nowcasting, deep learning has better performance than traditional methods. The increase of lead time reduces the guidance of real echoes evolution to extrapolation. This leads to evolution information loss in deep learning-based
prediction. Existing models, including GAN models, are difficult to reduce the loss. The accuracy of rainfall prediction is insufficient, especially heavy rainfall. However, heavy rainfall prediction is important for disaster prevention. A spatiotemporal process intensification model based on GAN is designed to solve the problem.

## 3 Model

In GAN-argcPredNet v1.0, the generator generates predicted images according to input image sequences. Then, the predicted
and real images are fed into discriminator with dual channel input. The discriminator makes judgments and the parameters are updated by adversarial loss optimization. Adam is used as the optimizer, which is an extension of stochastic gradient descent (Kingma Diederik and Adam, 2014). The generator parameters are updated once every five times. GAN-argcPredNet v2.0 model is constructed based on GAN-argcPredNet v1.0.





### 3.1 GAN-argcPredNet v2.0

GAN-argcPredNet v2.0 consists of STIC-Prediction generator and CS-Convolution discriminator (Fig. 1). STIC-Prediction generator reduces information loss and echoes attenuation by intensifying the spatiotemporal variations of the previous feature sequence. The generator is composed of argcPredNet and STIC Attention module (Fig. 2). The argcPredNet is composed a series of repeatedly stacked modules, with a total of three layers (Zheng et al., 2022). Each layer of the module including the input convolutional layer ($A_l$), the recurrent representation layer ($R_l$), the prediction convolutional layer ($\hat{A}_l$)

and the error representation layer ($E_l$). $R_l$ learns image features and generates the feature map $R_l^T \in R^{H \times W \times C}$, where $H$, $W$ and $C$ denote layer, current prediction time, map height, map width and feature channel respectively. The feature map guides the lower layers to generate images. STIC Attention is after the second layer. The previous feature sequence, especially heavy rainfall feature, is intensified from the spatiotemporal dimension. Weights are assigned to different rainfall areas, which curbs the information loss and echoes attenuation. Then, the intensified $R_l^T$ is fed to the next layer for more accurate

image. The calculation method of STIC-Prediction is:

$$A_l^T = \begin{cases} x_T & if\ l = 0 \\ MAXPOOL(\gamma(f(E_{l-1}^T))) & 0 < l < L \end{cases}, \tag{1}$$

$$\hat{A}_l^T = \gamma(f(R_l^T)), \tag{2}$$

$$E_l^t = \left[\gamma(A_l^T - \hat{A}_l^T)\ ;\ \gamma(\hat{A}_l^T - A_l^T)\ \right], \tag{3}$$

$$R_l^t = \begin{cases} argcLSTM(E_l^{T-1},\ R_l^{T-1}) & if\ l = L \\ argcLSTM(E_l^{T-1},\ R_l^{T-1},\ UPSAMPLE(STIC(R_{l+1}^0 : R_{l+1}^T))) & if\ l = 1 \\ argcLSTM(E_l^{T-1},\ R_l^{T-1},\ UPSAMPLE(R_{l+1}^T)) & l = 0\ and\ 1 < l < L \end{cases} \tag{4}$$

Here, $x_T$ denotes the initial input, $MAXPOOL$ denotes the maximum pooling operation, $\gamma$ denotes relu activation function, $f$ denotes convolution operation, $argcLSTM$ denotes Advanced Reduced-Gate Convolutional LSTM (Zheng et al., 2022), $STIC$ denotes STIC Attention.

CS-Convolution discriminator is composed of four-layer convolution structure and CS Attention module. Convolution structure is responsible for extracting echo features of input radar echo images. CS Attention module is embedded after the

first layer convolution structure. The module strengthens spatial information of echo features, especially heavy rainfall, which enhances discriminant ability. Then, the guidance is better given to the generator.



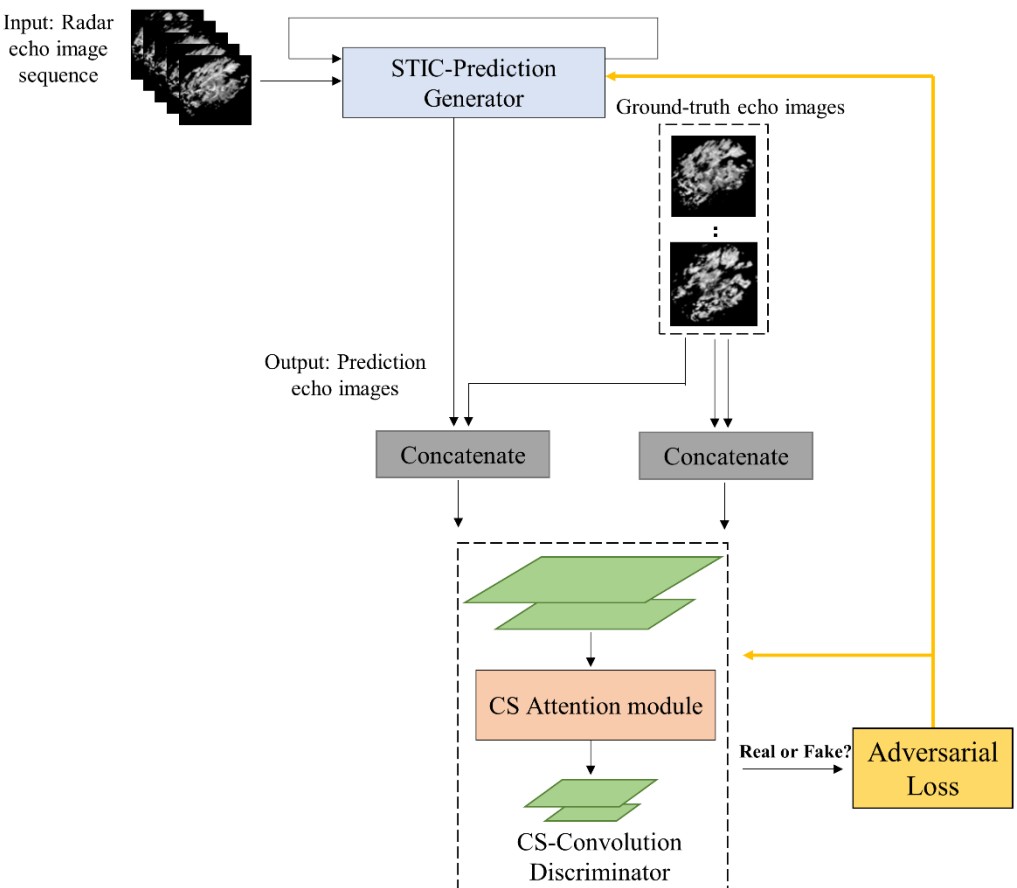

**Figure 1: This is the structure of GAN-argcPredNet v2.0. Twelve images are used in the sequence of radar echo maps that form the set of predictors. Here, five images are used as the input sequence, and seven images are used as the ground-truth images.**





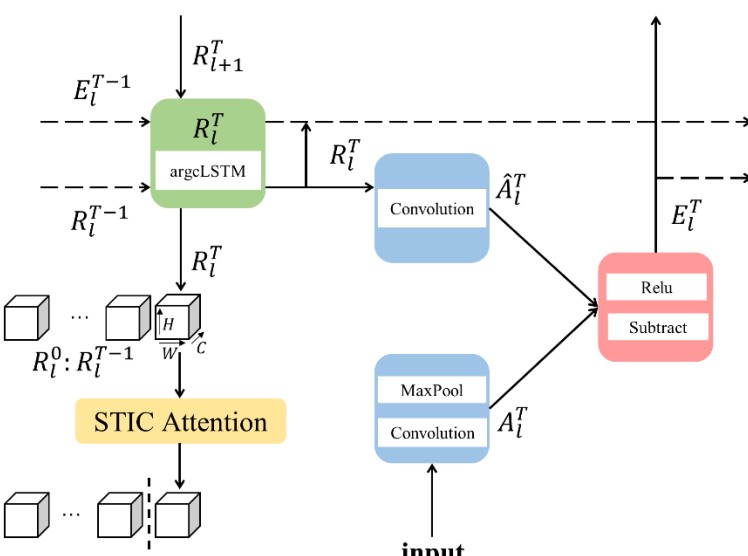


**Figure 2: This is the structure of STIC-Prediction. Modules are in layer 1 at time $T$.**

### 3.2 Intensified Spatiotemporal Evolution Information of Echoes

During extrapolation, the increase of lead time leads to a weaker correlation between real rainfall evolution and generated images. The evolution information is easily lost, which results in echoes attenuation. Intensifying the influence of previous
evolution can effectively solve the problem. STIC Attention intensifies spatiotemporal evolution information of echo feature sequences, which curbs the information loss and echoes attenuation. CS Attention is designed to improve the discriminant ability of rainfall by intensifying spatial information of echo features.

### 3.2.1 STIC Attention

The STIC Attention module is used in generator (Fig. 3). The STIC Attention combines MaxPool3D (3D = map height, map
width and time) and AvgPool3D to focus on the spatial information of feature sequences from the maximum and average perspectives. It focuses on heavy rainfall echoes, while considering non-heavy rainfall. With the introduction of the 3D convolution, the spatiotemporal changes of feature sequences are extracted. The representation ability of generator is enhanced from the spatiotemporal dimensions. Following are the detailed steps:

Given the feature sequence $F \in R^{t \times H \times W \times C}$ as input, where $t$ denotes time. Two feature sequences $F_{max}^{ts} \in R^{t \times H \times W \times 1}$ and
$F_{avg}^{ts} \in R^{t \times H \times W \times 1}$, are obtained by pooling operation, which denote the maximum and average feature along the channel axis respectively. The feature sequences are then connected and 3D convoluted. By using hard_sigmoid as the activation function, the training speed is accelerated. Then, the STIC Attention map sequence $M_{STIC} \in R^{t \times H \times W \times 1}$ is obtained. Finally, the output feature sequence $F_1 \in R^{t \times H \times W \times C}$, is calculated by element-wise multiplying $M_{STIC}$ and $F$. In short, the calculation method of





STIC Attention is:

$\quad M_{STIC} = \sigma(f_{7\times7\times5}((MaxPool3D(F))concat(AvgPool3D(F)))) = \sigma(f_{7\times7\times5}(F_{max}^{ts} concat F_{avg}^{ts}))$ , $\qquad$ (5)

$F_1 = M_{STIC} \otimes F$ , $\qquad$ (6)

Here, $concat$ denotes connection operation, $MaxPool3D$ denotes 3D maximum pooling operation, $AvgPool3D$ denotes 3D average pooling operation, $f_{7\times7\times5}$ denotes 3D convolution operation with convolution kernel of 7×7×5, and $\sigma$ denotes hard_sigmoid activation function, $\otimes$ denotes element-wise multiplication.

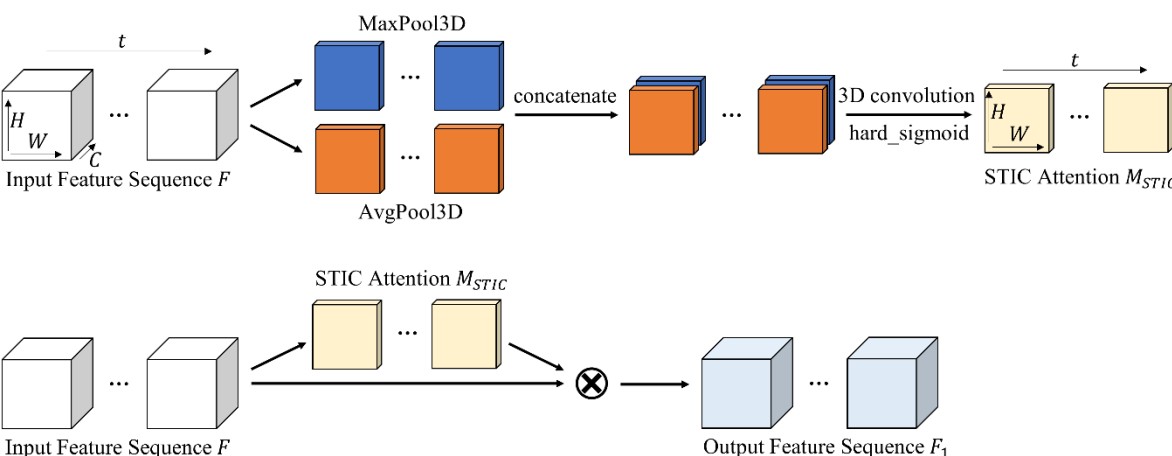


**Figure 3: This is the structure of STIC Attention.**

### 3.2.2 CS Attention

The CS Attention module (Fig. 4) is used in the discriminator. For input feature $F' \in R^{H \times W \times C}$, the channel attention map $M_c \in R^{1 \times 1 \times C}$, is generated by channel attention module. After element-wise multiplying with initial feature image, the

$\quad$ spatial attention map $M_s \in R^{H \times W \times 1}$, is generated by spatial attention module. Finally, the output feature $F'_2 \in R^{H \times W \times C}$ is obtained in the same way. In short, the calculation process is as follows:

$F'_1 = M_c \otimes F'$ , $\qquad$ (7)

$F'_2 = M_s \otimes F'_1$ , $\qquad$ (8)

Here, $\otimes$ denotes element-wise multiplication, $F'_1$ is the middle feature.





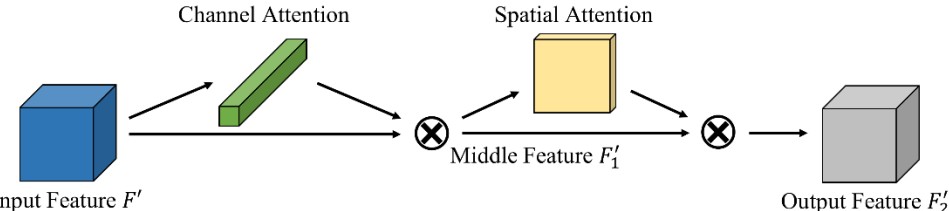

**Figure 4: This is the structure of CS Attention.**

The Channel Attention module (Fig. 5) studies relationship between the different feature channels. The global maximum and average pooling are used to gather spatial maximum and average information on each channel. The combination of them can judge the importance of different feature channels more comprehensively. Then the correlation between feature channels is obtained by learning the respective parameters in a dense layer. The Channel Attention assigns more weight to meaningful features from channel dimension. The detailed steps are as follows:

The feature map $F'$ is input into Channel Attention module. Two 1D feature maps $F_{max}^c \in R^{1 \times 1 \times C}$ and $F_{avg}^c \in R^{1 \times 1 \times C}$, are obtained by global pooling, which denote the global maximum and average pool feature respectively. Then, the correlation between features is extracted through dense layers. In order to reduce parameter overhead, the number of neurons in the first dense layer is set to $C/r$, where $r$ is the compression ratio. Then, the two results are summed. Finally, hard_sigmoid is used as the activation function to accelerate the training speed and obtain final channel attention map $M_c$. In short, the channel attention maps are calculated as follows:

$$M_c = \sigma(\varphi_{11}(\gamma(\varphi_{10}(GMP(F'))))+\varphi_{21}(\sigma(\varphi_{20}(GAP(F'))))) = \sigma(\varphi_{11}(\gamma(\varphi_{10}(F_{max}^c)))+\varphi_{21}(\sigma(\varphi_{20}(F_{avg}^c)))) , \quad (9)$$

Here, $GMP$ denotes global maximum pooling, $GAP$ denotes global average pooling, $\varphi_{10}$, $\varphi_{11}$ denotes the first and second dense layer of $F_{max}^c$, $\varphi_{20}$, $\varphi_{21}$ denotes the first and second dense layer of $F_{avg}^c$ and $\gamma$ denotes relu activation function.

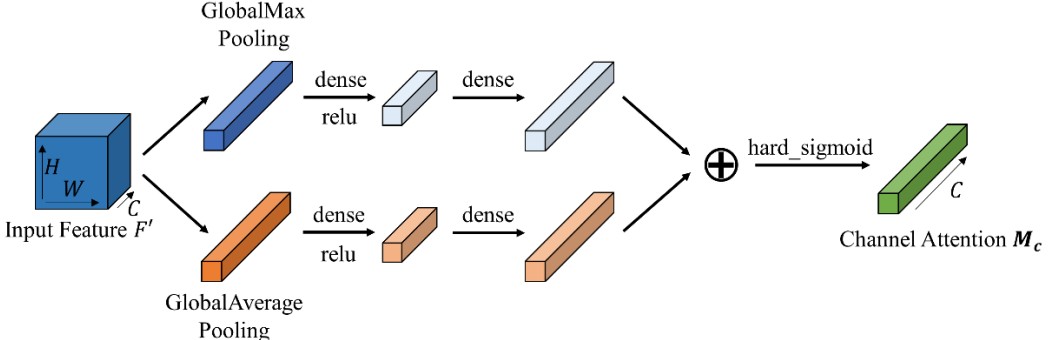

**Figure 5: This is the structure of Channel Attention.**

The Spatial Attention module (Fig. 6) studies the importance of each part in the same channel. The maximum and average pooling are used along channel axis respectively, which obtains echoes information of the feature image. The 2D convolution operation extracts feature and generates a spatial attention map with the same size as input image. The detailed steps are as follows:





After the Channel Attention module, the feature map $F_1' \in R^{H \times W \times C}$ is input into the Spatial Attention module. Two 2D feature maps $F_{max}^s \in R^{H \times W \times 1}$ and $F_{avg}^s \in R^{H \times W \times 1}$, are obtained by pooling operation, which denote the maximum pool feature and average pool feature on the channel respectively. The feature maps are then connected and 2D convoluted, using hard_sigmoid as the activation function to accelerate the training speed and obtain final spatial attention map $M_s$. In short, the calculation method of spatial attention map is:

$$M_s = \sigma(f_{7\times7}((MaxPool2D(F_1'))concat(AvgPool2D(F_1')))) = \sigma(f_{7\times7}(F_{max}^s concat F_{avg}^s)), \tag{10}$$

Here, $MaxPool2D$ denotes 2D maximum pooling operation, $AvgPool2D$ denotes 2D average pooling operation, amd $f_{7\times7}$ denotes 2D convolution operation with convolution kernel of 7×7.

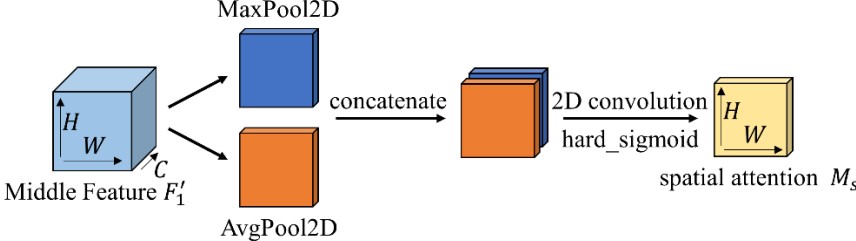

**Figure 6: This is the structure of Spatial Attention.**

## 4 Experiments and discussion

In South China radar echo data, continuous rainfall processes are selected as the training set from 2015 to 2016. The continuous radar echo images from March to May 2017 are selected as the testing set. The comparison with other models, e.g., ConvGRU, ConvLSTM, GA-ConvGRU, and GAN-argcPredNet v1.0, shows that GAN-argcPredNet v2.0 has better effect on rainfall prediction.

### 4.1 Dataset description

The paper uses the South China radar echo data provided by Guangzhou Meteorological Administration. The radar mosaic comes from 11 weather radars. The median filtering algorithm is used to control radar data quality, which eliminates errors caused by isolated clutter. In addition, the mirror filling and continuity checks are applied to remove traditional radar error sources. After quality control, there is only an extremely small amount of strong interference, which has negligible impact on the training of the model.

From 2015 to 2016, 32,004 consecutive echo images with rainfall are randomly selected as the training set. 7,992 consecutive images are randomly selected for testing from March to May 2017. The original resolution of the image is 1050 × 880, and each image covers 1050 km × 880 km. The pixel denotes the resolution of 1 km × 1 km. The reflectivity range is 0-80 dBZ, and the amplitude limit is between 0 and 255. The data is collected every 6 minutes, with the height of 1 km. To speed up the training and reduce the hardware load, the central 128 × 128 images are segmented.





## 4.2 Evaluation Metrics

As for evaluation, the paper uses five metrics to evaluate the prediction accuracy of all $128 \times 128$ pixels, which are
Probability of Detection (POD), False Alarm Ratio (FAR), Critical Success Index (CSI) and Heidke Skill Score (HSS). The
formulas for calculating these five indicators are as follows:

$$POD = \frac{TP}{TP+FN} , \tag{11}$$

$$FAR = \frac{FP}{TP+FP} , \tag{12}$$

$$CSI = \frac{TP}{TP+FN+FP} , \tag{13}$$

$$HSS = \frac{2(TP \times TN - FN \times FP)}{(TP+FN)(FN+TN)+(TP+FP)(FP+TN)} , \tag{14}$$

Here, $TP$ denotes that the real and predicted value reach specified threshold, $FN$ denotes that the real value reaches the specified threshold and the predicted value does not reach, $FP$ denotes that the real value does not reach specified threshold and the predicted value reaches, and $TN$ denotes that the real value and predicted value do not reach specified threshold.
POD evaluates hit ability and FAR is the indicator of false alarms. The combination of them can evaluate the model more
objectively. CSI and HSS are two composite metrics that provide a direct judgment of model effectiveness. The full score of
POD, CSI and HSS is 1. The full score of FAR is 0.

## 4.3 Training Setting

Before training, each pixel of the radar echo image is normalized to [0, 1]. All experiments are implemented by Python.
Model training and testing based on the Keras deep learning library with Tensorflow as backend. The operating environment
is a Linux workstation equipped with NVDIA RTX 2080 Ti 11G GPU.

## 4.4 Experiment results

Different thresholds of rainfall are set in the experiment, which are 0.5mm h[-1], 2mm h[-1], 5mm h[-1], 10mm h[-1] and 30mm h[-1]
respectively. Due to the relationship between radar reflectivity and rainfall type (Watters et al., 2021), the value on the radar
echo image is converted to the corresponding rainfall. The calculation formula is as follows:

$$Z = 10 \log a + 10b \log R , \tag{15}$$

Here, $a$ is set to 58.53 and $b$ is set to 1.56, $Z$ denotes radar reflectivity intensity, $R$ denotes rainfall intensity. The
correspondence between rainfall, rainfall intensity and rainfall level are referred to Table 1 (Shi et al., 2017).

**Table 1: This is the rainfall level table.**





| Rain Rate (mm h$^{-1}$) | Radar Reflectivity Intensity (dBZ) | Rainfall Level |
|---|---|---|
| $0 \leq R < 0.5$ | $Z < 12.98$ | No / Hardly noticeable |
| $0.5 \leq R < 2$ | $12.98 \leq Z < 22.37$ | Light |
| $2 \leq R < 5$ | $22.37 \leq Z < 28.58$ | Light to moderate |
| $5 \leq R < 10$ | $28.58 \leq Z < 33.27$ | Moderate |
| $10 \leq R < 30$ | $33.27 \leq Z < 40.72$ | Moderate to heavy |
| $30 \leq R$ | $40.72 \leq Z$ | Rainstorm warning |

In this paper, GAN-argcPredNet v2.0 is compared with other models. They are ConvGRU, ConvLSTM, GA-ConvGRU and
GAN-argcPredNet v1.0, respectively. The first five are common models in radar echo extrapolation, and the last one is the
model we designed before. From Fig. 7, 8, 9, 10 and 11, the POD, FAR, CSI and HSS score curves of GAN-argcPredNet
v2.0 are mostly in the best position during extrapolation.

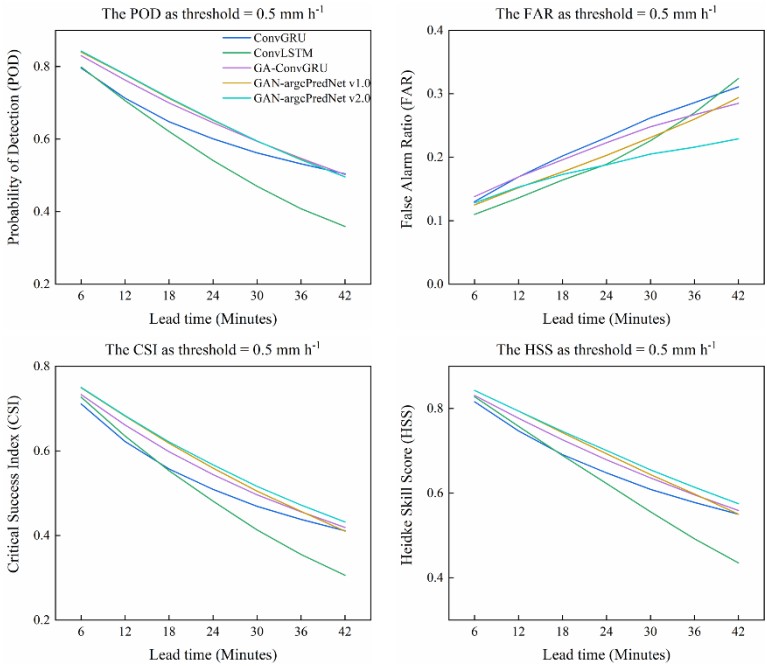

**Figure 7: This is the scores of POD, FAR, CSI and HSS under different radar extrapolation lead time when the**
**threshold = 0.5 mm h$^{-1}$.**



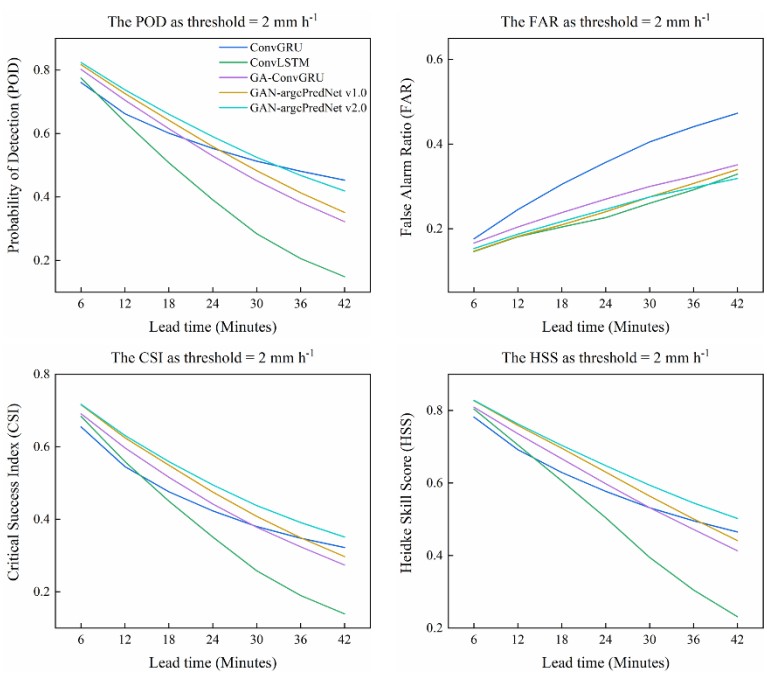

**Figure 8: This is the scores of POD, FAR, CSI and HSS under different radar extrapolation lead time when the threshold = 2 mm h⁻¹.**

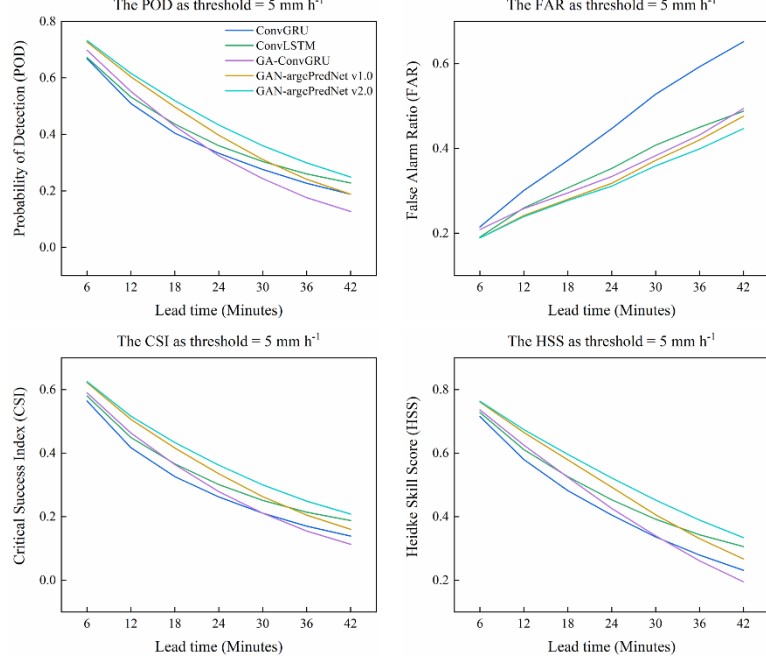

**Figure 9: This is the scores of POD, FAR, CSI and HSS under different radar extrapolation lead time when the threshold = 5 mm h⁻¹.**





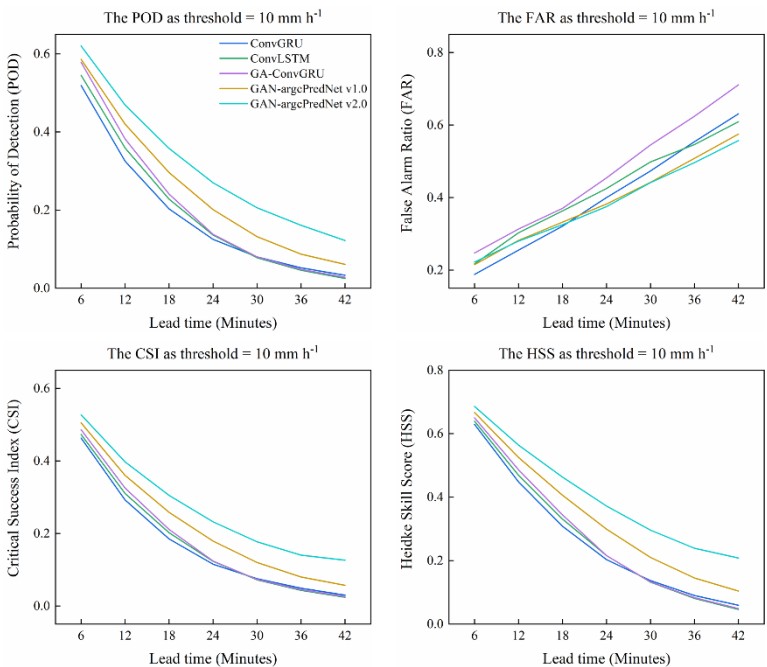

**Figure 10: This is the scores of POD, FAR, CSI and HSS under different radar extrapolation lead time when the threshold = 10 mm h⁻¹.**

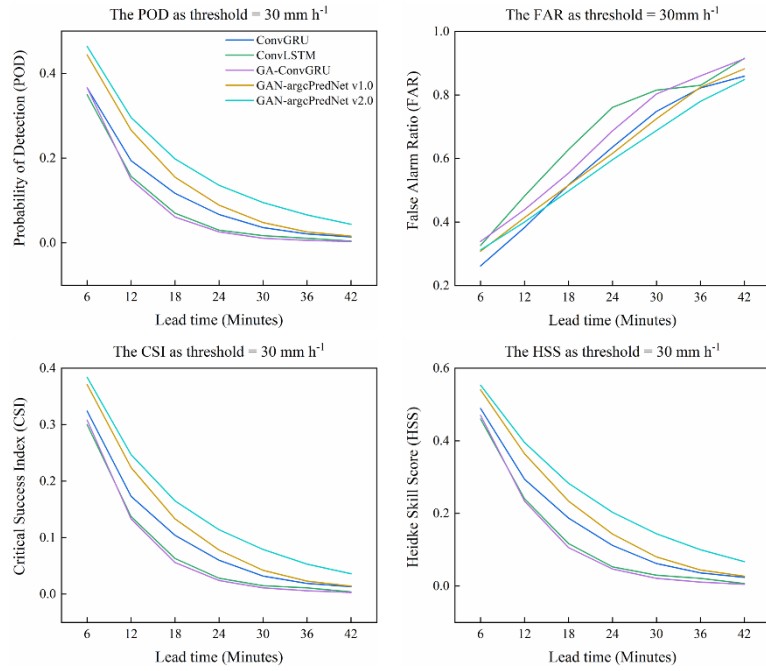


**Figure 11: This is the scores of POD, FAR, CSI and HSS under different radar extrapolation lead time when the threshold = 30 mm h⁻¹.**





In Table 2, the scores are the average performance of all lead times. Under different thresholds, the comprehensive performance of GAN-argcPredNet v2.0 exceeds other models. Especially in the heavy rainfall stages, the scores are far
better than others.

Using GAN-argcPredNet v1.0 as baseline, the POD, CSI and HSS scores of GAN-argcPredNet v2.0 increase by 23.5 %, 20.6 % and 19.0 % when the threshold is 10 mm h$^{-1}$. The FAR score also decreases by 1.5 %. When the threshold is 30 mm h$^{-1}$, the POD, CSI and HSS scores increase by 24.8 %, 22.2 % and 21.5 % respectively. The FAR score decreases by 3.76 %. GAN-argcPredNet v2.0 shows excellent performance in heavy rainfall prediction.

**Table 2: This is the average scores of POD, FAR, CSI and HSS under different threshold. Bold represents the best score.**

| Model | Threshold = 0.5 mm h$^{-1}$ | | | | Threshold = 2 mm h$^{-1}$ | | | |
|---|---|---|---|---|---|---|---|---|
|  | POD | FAR | CSI | HSS | POD | FAR | CSI | HSS |
| ConvGRU | 0.622 | 0.227 | 0.531 | 0.663 | 0.575 | 0.343 | 0.450 | 0.596 |
| ConvLSTM | 0.558 | 0.203 | 0.496 | 0.626 | 0.421 | **0.234** | 0.376 | 0.507 |
| GA-ConvGRU | 0.654 | 0.218 | 0.558 | 0.686 | 0.544 | 0.265 | 0.460 | 0.604 |
| GAN-argcPredNet v1.0 | 0.659 | 0.206 | 0.569 | 0.695 | 0.57 | 0.243 | 0.488 | 0.631 |
| GAN-argcPredNet v2.0 | **0.661** | **0.185** | **0.577** | **0.704** | **0.604** | 0.241 | **0.512** | **0.655** |

| Model | Threshold = 5 mm h$^{-1}$ | | | | Threshold = 10 mm h$^{-1}$ | | | |
|---|---|---|---|---|---|---|---|---|
|  | POD | FAR | CSI | HSS | POD | FAR | CSI | HSS |
| ConvGRU | 0.372 | 0.444 | 0.299 | 0.432 | 0.191 | 0.403 | 0.173 | 0.268 |
| ConvLSTM | 0.399 | 0.351 | 0.335 | 0.480 | 0.202 | 0.423 | 0.178 | 0.273 |
| GA-ConvGRU | 0.364 | 0.344 | 0.311 | 0.444 | 0.214 | 0.466 | 0.184 | 0.280 |
| GAN-argcPredNet v1.0 | 0.423 | 0.328 | 0.358 | 0.500 | 0.255 | 0.391 | 0.223 | 0.337 |
| GAN-argcPredNet v2.0 | **0.458** | **0.317** | **0.385** | **0.533** | **0.315** | **0.385** | **0.269** | **0.401** |

| Model | Threshold = 30 mm h$^{-1}$ | | | |
|---|---|---|---|---|
|  | POD | FAR | CSI | HSS |
| ConvGRU | 0.116 | 0.604 | 0.104 | 0.172 |
| ConvLSTM | 0.091 | 0.680 | 0.080 | 0.133 |





| | | | | |
|---|---|---|---|---|
| GA-ConvGRU | 0.089 | 0.657 | 0.077 | 0.128 |
| GAN-argcPredNet v1.0 | 0.149 | 0.612 | 0.126 | 0.205 |
| GAN-argcPredNet v2.0 | **0.186** | **0.589** | **0.154** | **0.249** |

The prediction examples are visualized in Fig. 12. Compared with other models, GAN-argcPredNet v2.0 has better predictions and the heavy rainfall echoes are more consistent with Ground-truth. These indicate that GAN-argcPredNet v2.0 effectively retains process information and curbs the attenuation.

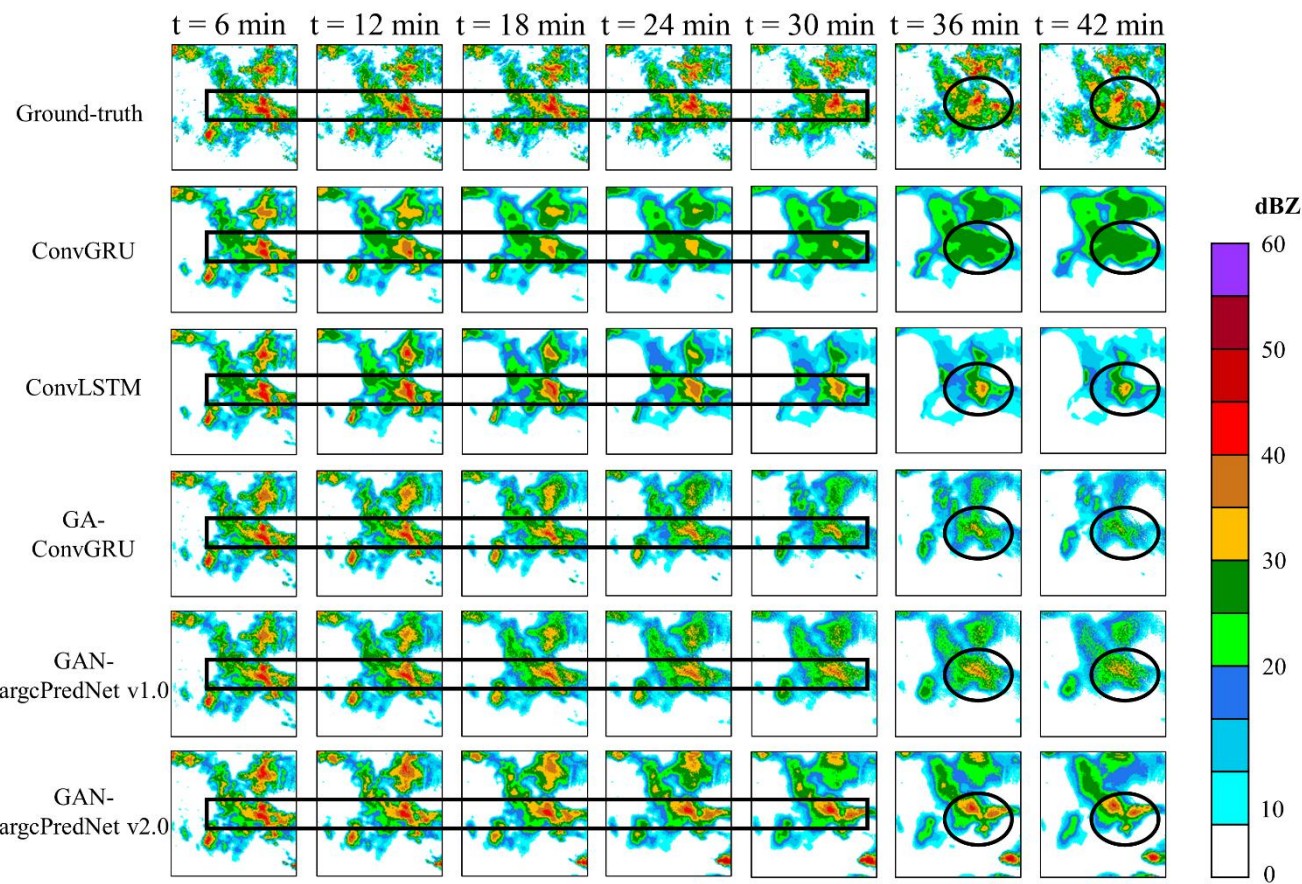


**Figure 12: This is the example of radar echo exploration. The circular and rectangular regions represent heavy rainfall prediction.**

In order to evaluate the quality of generated images objectively, Mean Square Error (MSE) and Mean Structural Similarity (MSSIM) are chosen for the experiment (Wang et al., 2004; Inoue and Misumi, 2022). The full score of MSE is 0 and

MSSIM is 1. According to Table 3, GAN-based models generate higher quality images, with GAN-argcPredNet v2.0 performing the best.

**Table 3: This is the MSE and MSSIM scores of each model. Bold represents the best score.**





| Model | MSE×$10^2$ | MSSIM |
|---|---|---|
| ConvGRU | 0.406 | 0.724 |
| ConvLSTM | 0.218 | 0.784 |
| GA-ConvGRU | 0.188 | 0.812 |
| GAN-argcPredNet v1.0 | 0.191 | 0.814 |
| GAN-argcPredNet v2.0 | **0.187** | **0.826** |

## 4.5 Ablation Study

Through ablation study, we investigate the effects of STIC Attention and CS Attention. STIC-GAN and CS-GAN are
constructed by adding STIC Attention module only in generator and CS Attention module only in discriminator. GAN-
argcPredNet v1.0, STIC-GAN, CS-GAN and GAN-argcPredNet v2.0 are tested respectively. From Fig. 13, 14, 15, 16 and 17,
CS-GAN and STIC-GAN perform better than GAN-argcPredNet v1.0.

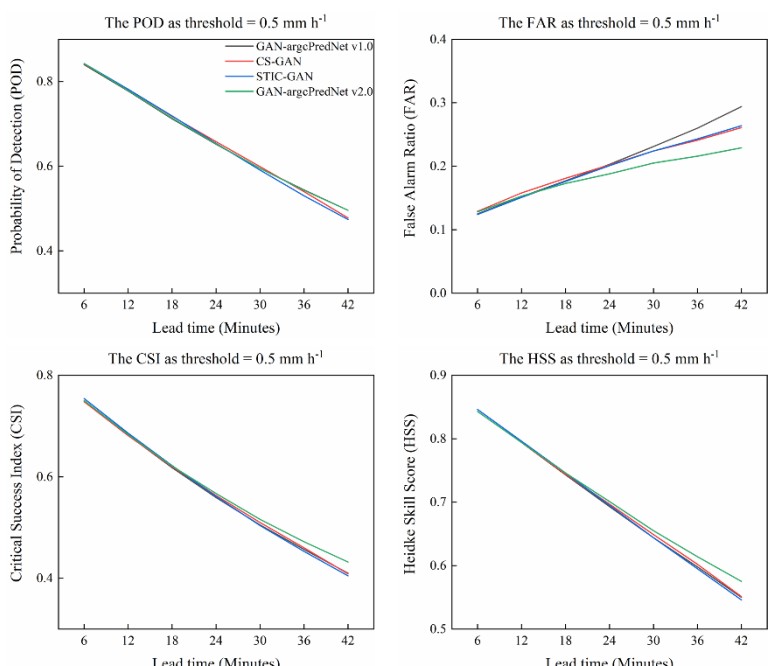

**Figure 13: This is the scores of POD, FAR, CSI and HSS under different radar extrapolation lead time when the**
**threshold = 0.5 mm h⁻¹.**





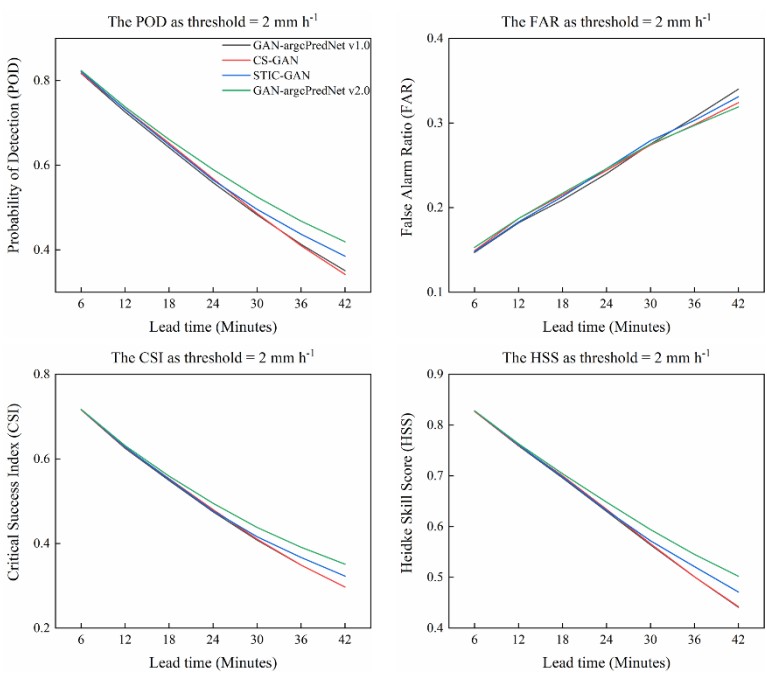

**Figure 14: This is the scores of POD, FAR, CSI and HSS under different radar extrapolation lead time when the threshold = 2 mm h⁻¹.**

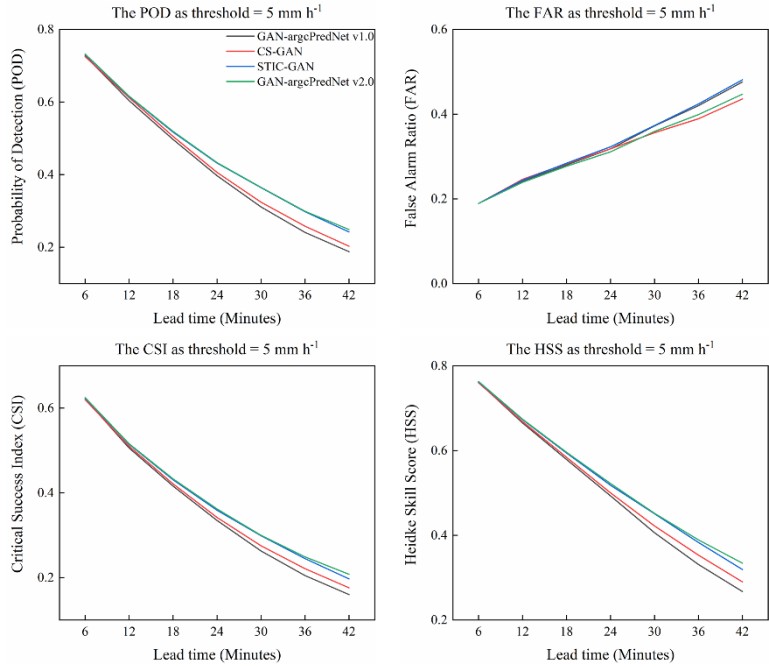

**Figure 15: This is the scores of POD, FAR, CSI and HSS under different radar extrapolation lead time when the threshold = 5 mm h⁻¹.**





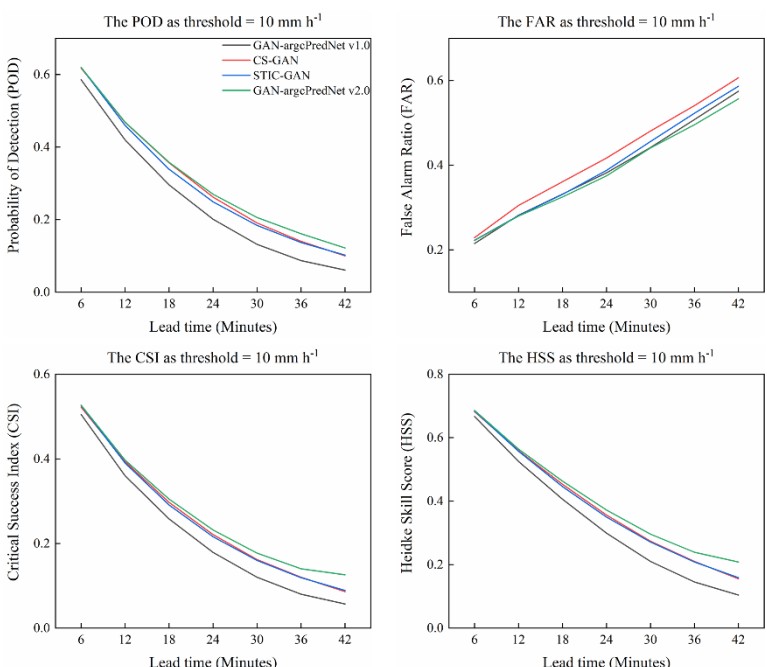

**Figure 16: This is the scores of POD, FAR, CSI and HSS under different radar extrapolation lead time when the threshold = 10 mm h⁻¹.**

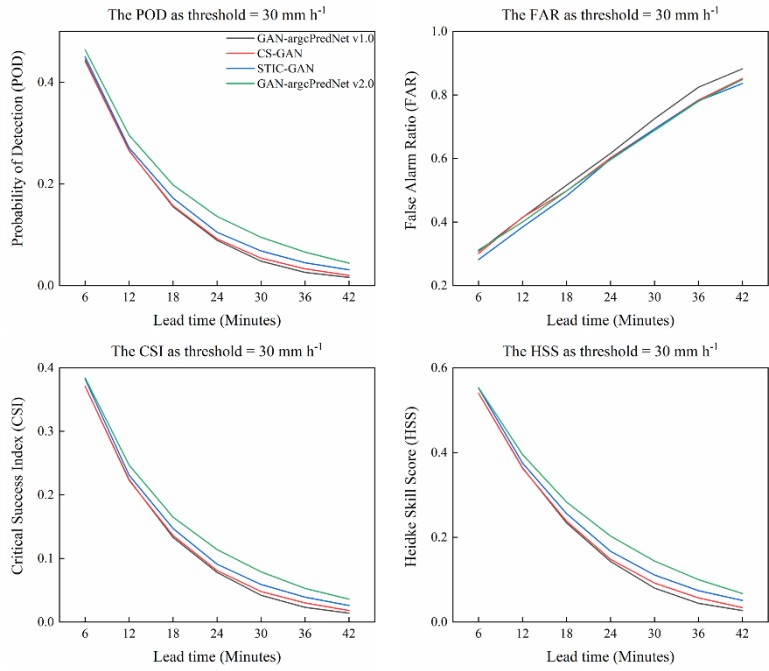


**Figure 17: This is the scores of POD, FAR, CSI and HSS under different radar extrapolation lead time when the threshold = 30 mm h⁻¹.**





From Table 4, when the threshold increases, the GAN-argcPredNet v2.0, CS-GAN and STIC-GAN far exceed the GAN-argcPredNet v1.0, and the STIC-GAN is closer to the GAN-argcPredNet v2.0. The STIC Attention focuses more on the more rapidly evolving areas of the radar maps, which leads to the result. The CS Attention makes GAN-argcPredNet v2.0 have better comprehensive performance.

**Table 4: This is the average scores of POD, FAR, CSI and HSS under different threshold. Bold represents the best score.**

| Model | Threshold = 0.5 mm h⁻¹ | | | | Threshold = 2 mm h⁻¹ | | | |
|---|---|---|---|---|---|---|---|---|
| | POD | FAR | CSI | HSS | POD | FAR | CSI | HSS |
| GAN-argcPredNet v1.0 | 0.659 | 0.206 | 0.569 | 0.695 | 0.57 | 0.243 | 0.488 | 0.631 |
| CS-GAN | 0.660 | 0.200 | 0.571 | 0.697 | 0.573 | 0.242 | 0.490 | 0.632 |
| STIC-GAN | 0.656 | 0.198 | 0.570 | 0.696 | 0.584 | 0.243 | 0.498 | 0.641 |
| GAN-argcPredNet v2.0 | **0.661** | **0.185** | **0.577** | **0.704** | **0.604** | **0.241** | **0.512** | **0.655** |

| Model | Threshold = 5 mm h⁻¹ | | | | Threshold = 10 mm h⁻¹ | | | |
|---|---|---|---|---|---|---|---|---|
| | POD | FAR | CSI | HSS | POD | FAR | CSI | HSS |
| GAN-argcPredNet v1.0 | 0.423 | 0.328 | 0.358 | 0.500 | 0.255 | 0.391 | 0.223 | 0.337 |
| CS-GAN | 0.433 | 0.317 | 0.366 | 0.511 | 0.306 | 0.421 | 0.257 | 0.384 |
| STIC-GAN | 0.458 | 0.331 | 0.382 | 0.529 | 0.299 | 0.397 | 0.256 | 0.383 |
| GAN-argcPredNetv2.0 | **0.458** | **0.317** | **0.385** | **0.533** | **0.315** | **0.385** | **0.269** | **0.401** |

| Model | Threshold = 30 mm h⁻¹ | | | |
|---|---|---|---|---|
| | POD | FAR | CSI | HSS |
| GAN-argcPredNet v1.0 | 0.149 | 0.612 | 0.126 | 0.205 |
| CS-GAN | 0.152 | 0.592 | 0.129 | 0.210 |
| STIC-GAN | 0.163 | **0.580** | 0.139 | 0.226 |
| GAN-argcPredNet v2.0 | **0.186** | 0.589 | **0.154** | **0.249** |

The effects of STIC Attention and CS Attention on extrapolation is shown in Fig. 18. The addition of STIC Attention or CS Attention makes the reflectivity of heavy rainfall echoes closer to the real echoes. Rainfall information is still well



maintained in the seventh images. The combination of the two makes the effect more obvious. Intensifying the influence of the previously input rainfall evolution information reduces echoes attenuation.

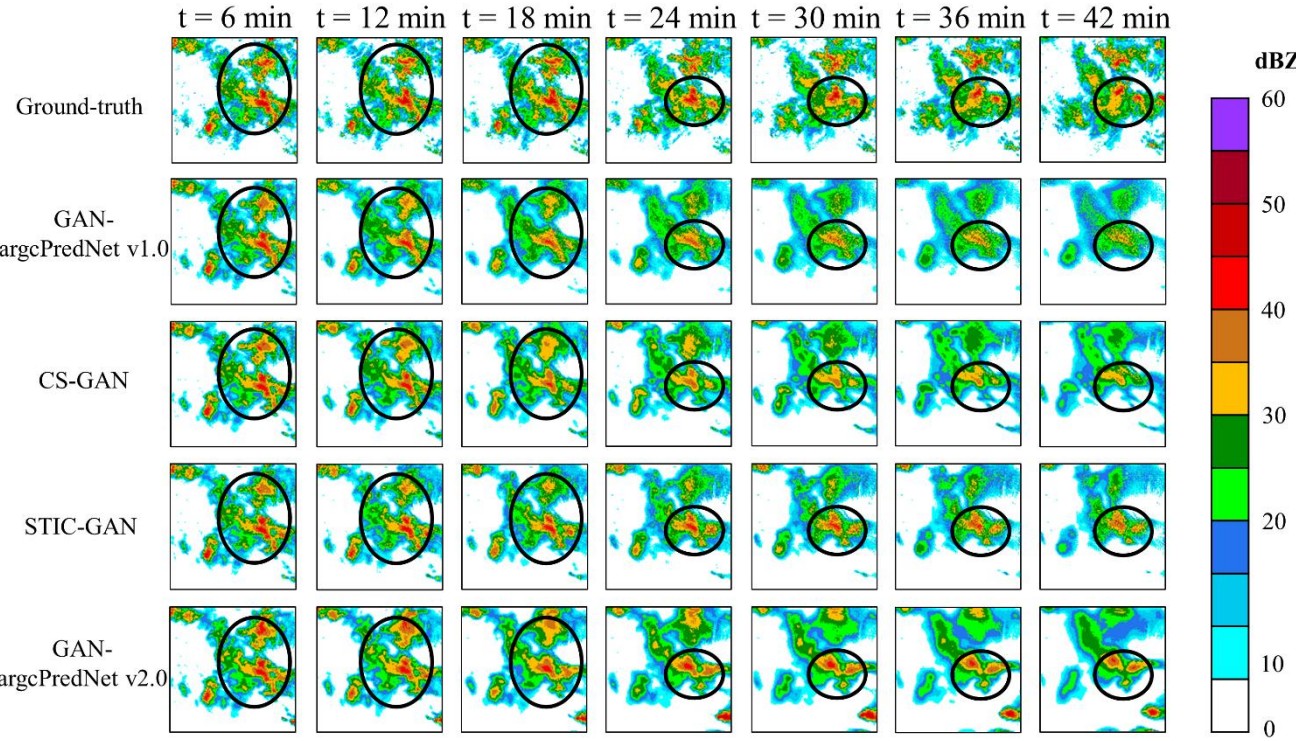

**Figure 18: This is the example of radar echo exploration. The circular regions represent heavy rainfall prediction.**

## 330    5 Conclusion

The study improves precipitation nowcasting by reducing information loss and echoes attenuation. With the intensification of the previously input rainfall evolution information, GAN-argcPredNet v2.0 reduces information loss and enhances the prediction accuracy of rainfall, especially heavy rainfall. Meanwhile, the model is designed based on the generative adversarial structure, which achieves high quality radar echo extrapolation.

In practice, a predictive software has been developed based on our model. After the software accesses the radar data and establishes a prediction task, rainfall prediction results are output as dataset. Then the dataset can be fed into the urban flood warning system. The improvement of rainfall prediction has a positive impact on flood prediction and urban-operation safety. Overall, GAN-argcPredNet v2.0 is a spatiotemporal process intensification model based on GAN, which achieves more accurate rainfall prediction. However, there are further improvements on the basis of current accuracy.



Future work should also consider how to achieve high-resolution rainfall prediction. High-resolution prediction is often limited by hardware conditions. Therefore, further optimization of the model to attenuate hardware conditions required for training and predicting is a realistic research direction.

**Code and data availability**

The radar data used in the paper comes from Guangdong Meteorological Administration. Due to the confidentiality policy,
we only provide a sequence of 12 images. If you need to access more data, please contact Kun Zheng (ZhengK@cug.edu.cn) and Qiya Tan (ses_tqy@cug.edu.cn). The GAN-argcPredNet v2.0 model is open source. You can find the source code from https://doi.org/10.5281/zenodo.7505030.

**Author contributions**

Kun Zheng and Qiya Tan were responsible for developing models and writing manuscripts; Huihua Ruan, Jinbiao Zhang,
Cong Luo, Siyu Tang Yunlei Yi, Yugang Tian and Jianmei Cheng were responsible for data screening and preprocessing.

**Competing interests**

The authors declare that they have no conflict of interest.

**Acknowledgements**

This work was supported by Science and Technology Planning Project of Guangdong Province, China [grant
No.2018B020207012].

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
