# Peer review of "GAN-argcPredNet v2.0: A Radar Echo Extrapolation Model based on Spatiotemporal Process Enhancement"

_Geoscientific Model Development, 2022_

## Author Comment (AC1)

**RESPONSE TO REVIEWER'S COMMENTS**

We would like to thank the reviewer's valuable suggestions for our manuscript. The corrections and suggestions made by the reviewer improved our manuscript. The comments from the reviewer also provide us with future research directions.

*Comments from Reviewer 1:*

*Comment 1: The structure of the manuscript is not well organized and redundant. For example, the last two sentences of the introduction section (i.e., Section 1) mention about the experimental setup and the results. They should be in the dedicated sections for the "experimental setup" and "results." The first paragraph of Section 2 also repeats the same thing as that in Section 1.*

**Response:** Thanks for your suggestion. We have re-structured the manuscript. The current structure of the manuscript consists of Introduction, Related Work, Model, Data and experimental Setup, Results, Discussions and conclusions. The redundant sentences have also been deleted (Page 2, Line 59-62; Page 3, Line 64-66; Page 4, Line 116-121; Page 7, Line 160-165; Page 10, Line 233-236).

*Comment 2: Line 126, "STIC-Prediction generator reduces information loss...": This is not proven yet up to this line in the manuscript, nor supported by previous studies (indeed, no citation here). Therefore, this is just your working hypothesis at this moment. If you intend to prove that STIC-Prediction generator reduces information loss, it should be in the results section, not here. Besides, you need to define a metric for information loss. Otherwise, you cannot prove it quantitatively. I would recommend writing as follows: "STIC-Prediction generator is designed to reduce information loss and...". Other sentences in the same paragraph have the same problem. Please clearly separate introduction, experimental setup, result, and discussion.*

**Response:** Thanks for your advice. We have modified the sentences and added the Bias metric to measure information loss (Page 5, Line 130; Page 5, Line 136-140; Page 6, Line 151-154; Page 11, Line 260). The structure of the manuscript is now Introduction, Related Work, Model, Data and experimental Setup, Results, Discussions and conclusions.

*Comment 3: Line 153-157: This is the same as the previous comment. This is your working hypothesis until you present evidence for that.*

**Response:** Thanks for your advice. We have deleted the sentences because they are repeated in Page 2, Line 43, Page 5, Line 138, and Page 6, Line 151. You can find the deletion on Page 7, Line 160-165.

**Comment 4:** *Line 166, "By using hard_sigmoid as the activation function, the training speed is accelerated.": This is interesting, but not supported by evidence. You may accelerate the training near the origin, where the gradient is non-zero, while you will decelerate the training where the gradient gets exactly zero.*

**Response:** Thanks for your advice. The construction of hard_sigmoid allows gradients to flow easily when the unit is not saturated, while providing a crisp decision in the saturated regime. Therefore, it has lower computational cost and faster calculation speed. This is the reason why we chose hard_sigmoid. The sentences in the manuscript may have caused ambiguity, so we have modified them. We have also cited relevant references that mention this phenomenon (Page 8, Line 177).

**Comment 5:** *Page 8-10: In general, descriptions lack important information, such as the number of channels, padding, stride, batch size, etc. All these hyperparameters affect the model performance. Comparing different models without knowing setup of each model does not really make sense. At least, you should provide details of all the models you used in this paper. If it is too big, you can put them in the appendix or supporting information.*

**Response:** Thanks for your suggestion. We have provided the hyperparameters of ConvLSTM, ConvGRU, GA-ConvGRU, GAN-argcPredNet v1.0 and GAN-argcPredNet v2.0 in the supplement.

**Comment 6:** *Line 219-221: It is a little bit strange to conclude before showing results.*

**Response:** Thanks for your suggestion. We have deleted the sentences because they are repeated in Page 10, Line 238, Line 243 and Page12, Line 280. You can find the deletion on Page 10, Line 233-236.

**Comment 7:** *Section 4.1-4.3: They are the experimental setup. I would recommend to separate the experimental setup from the result section.*

**Response:** Thanks for your advice. We have separated the experimental setup from the result section (Section 4 and Section 5). The structure of the manuscript is now Introduction, Related Work, Model, Data and experimental Setup, Results, Discussions and conclusions.

**Comment 8:** *Equation 15: This is somewhat strange. Usually, radar reflectivity is converted to rain rate with the equation as follows (e.g., https://glossary.ametsoc.org/wiki/Z-r_relation):*
*$Z = a * R^b$.*
*Then, the radar reflectivity is expressed in decibel (dBZ, https://glossary.ametsoc.org/wiki/Dbz):*
*$dBZ = 10 * \log10(Z)$.*
*Although Eq 15 is the same as that in Shi et al. (2017), your definition uses log, not log10. If that is the case, your rain rate may be wrong. Please check your code and reprocess data if necessary.*

*Despite the wrong definition, the values in Table 1 seem correct. In addition, the citation just before this equation (Watters et al. 2021) seems misplaced. I do not understand why this is cited here.*

**Response:** Thanks for pointing it. The calculation in the code is based on $log_{10}$. We have modified the formula and revised the issues in the manuscript writing (Page 11, Line 248; Page 11, Line 250).

*Comment 9: Line 261, "From Fig. 7, 8, 9, 10, and 11": Although you placed 5 figures with 4 panels each in the manuscript, you just describe them in a single line. This is not acceptable. Please add meaningful descriptions and discussion for them, or please consider reducing the figures.*

**Response:** Thanks for your advice. We have added descriptions and discussions of the figures, and simplified the original figures (Page 12, Line 289-300; Section 6.1).

*Comment 10: Figures 7-11, 13-17: These figures have 4-5 lines each, but they are not clear, and they are drawn with similar colors (many of them are in blueish colors). Please improve their quality. However, before improving them, please consider summarizing them more concisely.*

**Response:** Thanks for your advice. We have summarized them more concisely and changed the colors, symbols, and styles of lines to make the figures clearer. In the ablation study, we have replaced the figure with the table to present the data more specifically (Page 13, Line 299; Page 15, Line 324).

*Comment 11: Figure 12: You highlighted an intense rain area near the center of the domain, which is predicted well by the proposed method. Meanwhile, the rain area on the bottom right corner over-intensifies in the proposed method. The rain area near the top right in the domain goes out of the domain, which is different from the ground truth. In other methods, this rain area is located at the right place. I would say that the proposed method has pros and cons. Please discuss the results more carefully. Not just saying that you make a great system, but a more scientific consideration is needed. I imagine that the over-intensification near the bottom right may be related to the use of attention mechanism.*

**Response:** Thanks for your suggestion. We have pointed out this issue in the manuscript and discussed it carefully (Page 13, Line 305; Page 16, Line 347-353).

*Comment 12: Line 20, "intensification of echo feature sequence": It is not clear what you intensified. If you always increase the extracted features by CNN, I do not think you can make a good prediction. I guess you meant the attention mechanism, but this sentence did not make sense.*

**Response:** Thanks for your suggestion. We have modified the sentence (Page 1, Line 20).

*Comment 13: Line 45, "Existing deep learning models, ...": It is impossible to prove non-existence, so I would recommend adding "to the knowledge of authors."*

**Response:** Thanks for your advice. We have modified the sentence (Page 2, Line 46).

***Comment 14:*** *Line 77, "these traditional methods fail to utilize": Most of the traditional methods are semi-process-based, so they do not intend to utilize historical data directly. Therefore, it is not fair to use the phrase "fail to."*

**Response:** Thanks for pointing it. We have modified the expression (Page 3, Line 81).

***Comment 15:*** *Line 130, "where H, W and C denote": "T" and "l" are missing.*

**Response:** Thanks for pointing it. We have fixed it (Page 5, Line 135).

***Comment 16:*** *Equation 9: \sigma before \phi_21 (there are two) must be \gamma.*

**Response:** Thanks for pointing it. We have modified the equation (Page 9, Line 213).

***Comment 17:*** *Table 4, the values for 5 mm/h, FAR: CS-GAN and GAN-argcPredNetv2.0 show the same number within significant digits. Therefore, both should be presented with bold.*

**Response:** Thanks for advice. We retrained and retested the data to extend the prediction time to 1 h. As a result, the numbers in the table have changed, but we carefully checked them this time (Page 15, Line 324).

***Comment 18:*** *Line 339, "However, there are further improvements on the basis of current accuracy.": I could not understand the sentence. Please consider rephrasing.*

**Response:** Thanks for pointing it. We have modified the expression (Page 17, Line 374).

***Comment 19:*** *"hardware conditions": What does this mean?*

**Response:** Thanks for question. We mean that high-resolution prediction requires high hardware requirements, such as graphics cards with larger memory. We have modified the expression (Page 17, Line 376).

---

## Author Comment (AC2)

**RESPONSE TO REVIEWER'S COMMENTS**

We would like to thank the reviewer's valuable suggestions for our manuscript. The corrections and suggestions made by the reviewer improved our manuscript. The comments from the reviewer also provide us with future research directions.

*Comments from Reviewer 2:*

**Comment 1:** *The written quality of this manuscript is poor. I recommend re-structuring the result section.*

**Response:** Thanks for your suggestion. We have separated the experimental setup and results, and re-structuring the result section (Section 5).

**Comment 2:** *Same as comment 1, more detailed descriptions for Fig. 7-11 need be included. In addition, in Figure 12, a general conclusion that the extrapolation of the new method is superior to other methods was given, the reason behind the results obtained in the manuscript should be provided. The false prediction showed in the lower right corner of the figure should be mentioned and explained.*

**Response:** Thanks for your advice. We have summarized the figures more concisely and provided a detailed description. We also have analyzed the reasons behind the results. The false predictions have been mentioned and explained (Page 12, Line 289-300; Page13, Line 304-306; Page 16, Line 341-345; Page 16, Line 347-353).

**Comment 3:** *Precipitation nowcasting is generally defined as the prediction within 0-2 hours, but in the manuscript, the extrapolation results for a longer time (such as 1h, or one hour later) are not mentioned and presented. please state this in the discussion part (that this work only focuses on the 45-minute prediction?).*

**Response:** Thanks for your advice. We have retrained and retested the data to achieve 1-hour extrapolation. The results and discussions have been updated (Section 5; Section 6.1).

**Comment 4:** *The description of input and output parameters of model training is too brief. Although the code is provided, more detailed model parameters should be listed.*

**Response:** Thanks for your suggestion. We have provided the hyperparameters of ConvLSTM, ConvGRU, GA-ConvGRU, GAN-argcPredNet v1.0, and GAN-argcPredNet v2.0 in the supplement.

***Comment 5:*** *The conclusion section of this manuscript is too brief. As a neural network-based study, many key concerns were not discussed. I recommend proposing a separate discussion section to summarize evaluation results, comparisons with other works.*

**Response:** Thanks for your suggestion. We have proposed a discussion section where we discuss and compare the model prediction results in detail, while analyzing the reasons (Section 6.1). The current structure of the manuscript consists of Introduction, Related Work, Model, Data and experimental Setup, Results, Discussions and conclusions.

---

## Author Response (AR2)

**RESPONSE LETTER**

We would like to thank the Topic editor and reviewers for the valuable suggestions on our manuscript once again. The suggestions made by the Topic editor and reviewers further improved our manuscript.

*Comments from Topic editor:*

**Comment 1:** *The authors emphasize that the new framework includes a module to intensify the spatiotemporal variations of the feature. However, the example they used did not justify this advantage. The example shown in Figure 8 exhibits decaying features during the last 30-min in the forecast. Although the GAN-argcPredNet V2.0 method alleviates the "attenuation" disadvantage shown in the results from other methods, it is unclear whether the decaying features are due to attenuation or well representation of the actual signals. I'd suggest the authors consider a different case (even better with multiple cases) to justify the advantage of the method, e.g. evident spatiotemporal variations with rapidly intensifying convection.*

**Response:** Thanks for your suggestion. We have added two prediction examples to more comprehensively demonstrate the advantages of our model. One example is the rapid growth of the storm, and the other example is the decay of the storm. Our model curbs echo attenuation in all examples and has the best performance (Page 15, Figure 10; Page 16, Figure 11).

**Comment 2:** *As pointed out by reviewer #1, the word "intensify" repeatedly appears. This word may need to be revised to describe the features the authors want to emphasize adequately. I'd suggest using other words, such as "enhance"?*

**Response:** Thanks for your suggestion. We have revised the description of "intensify" in the manuscript, taking into account your suggestions and those of reviewer #1 (Page 1, Line 18, Line 20; Page 3, Line 91-92; Page 4, Line 111; Page 5, Line 119-120; Page 18, Line 341-343; Page 19, Line 363-366). We have also changed the model name to " a Spatiotemporal Process Enhancement Network ".

**Comment 3:** *The authors did not explain why the strong echoes outside the center of the echo maps are not as well represented as the main features during the forecast. Please make some comments about this limitation in section 6.1.*

**Response:** Thanks for pointing it. We have added the relevant explanations in the manuscript (Page 19, Line 350-353). This phenomenon can be better explained by combining the previous descriptions (Page 3, Line 81-85).

***Comment 4:*** *For readers unfamiliar with the concept of generator and discriminator, these components should be briefly explained. Also, Figure 2 is provided with little explanation.*

**Response:** Thanks for your advice. We have briefly explained the role of generator and discriminator (Page 3, Line 69-72). We have also added descriptions in the figure caption (Page 7, Figure 3).

***Comment 5:*** *Caption in Figure 1: " Fifteen radar echo images are used in the testing set" Please clarify this.*

**Response:** Thanks for your advice. We have revised the sentence which was misleading (Page 6, Line 135-136).

*Comments from Reviewer 1:*

***Comment 1:*** *Page 1, line 20: "By intensifying ..." This wording is somewhat misleading. It is not proven in this manuscript whether the network always increases the "evolution." What is shown in this paper is that the new network implementation contributed to "suppress the blurring effect of rain distribution and reduce the negative bias (i.e., the Bias Score less than one)" by STIC Attention. Therefore, my suggestion is as follows:*
*"By suppressing the blurring effect of rain distribution and reducing the negative bias by STIC Attention, ..."*
*Throughout the manuscript, the authors emphasize that they "intensify" something. However, rain intensity grows or decays, following the lifecycle of precipitation systems. Therefore, the network should not intensify features regardless of environment. In addition, it is not explicitly proven whether the output of Attention blocks really increases the signals of hidden states or not. Scientific papers need to describe results objectively, not subjectively. Again, what can be read from the figures is that the network succeeded in suppressing the blurring effect and reducing the negative bias. You do not really need to introduce subjective interpretation on it.*

**Response:** Thanks for your suggestion. We have revised the description of "intensify" in the manuscript (Page 1, Line 18, Line 20; Page 3, Line 91-92; Page 4, Line 111; Page 5, Line 119-120; Page 18, Line 341-343; Page 19, Line 363-366).

***Comment 2:*** *Page 3, line 89: " In sequence prediction, temporal information is also important, but these methods fail to intensify it. For radar echo extrapolation, it reflects as a lack of intensification to rainfall evolution information." As in the comment above, the use of word "intensification" is not adequate. What is needed is to avoid blurring or to maintain the intensity.*

**Response:** Thanks for your advice. Since we have rewritten Section 1 and Section 2, this sentence has been removed. However, all other descriptions of "intensify" have been revised

(Page 1, Line 18, Line 20; Page 3, Line 91-92; Page 4, Line 111; Page 5, Line 119-120; Page 18, Line 341-343; Page 19, Line 363-366).

*Comment 3:* *Page 10, line 222: " To measure information loss, the paper also uses the Bias metric," The introduction of the Bias Score generally improved the discussion here. However, the Bias Score is a measure of bias. Bias does not necessarily reflect information loss. Therefore, I would suggest rephrasing the sentence as follows:*
*"To measure the blurring effect, the paper also uses the Bias Score."*
*Usually, the amount of information is measured by the Shanon entropy of information.*

**Response:** Thanks for pointing it. We have revised this sentence (Page 11, Line 232).

*Comment 4:* *Table S5-S8: These tables do not have information on padding. I still feel that information is not complete.*

**Response:** Thanks for pointing it. We have added the information on padding (Table S5-S8).

*Comment 5:* *Page 1, line 26: "Bias" This is not raw bias, but so-called "Bias Score."*

**Response:** Thanks for your advice. We have revised this description (Page 1, Line 25).

*Comment 6:* *Page 3, line 69: "The extrapolation accuracy is affected." This sentence is isolated and does not make sense. Please consider rephrasing.*

**Response:** Thanks for your advice. We have revised the sentence to make it more complete. (Page 2, Line 46).

*Comment 7:* *Page 3, line 77: "the natural variation motion" What does this mean? Please consider rephrasing.*

**Response:** Thanks for pointing it. We have revised the misrepresentation. (Page 2, Line 55).

*Comment 8:* *Figure 7: There is no information about the forecast time. Did you use all the time steps to compute the scores? If so, please describe it in the caption.*

**Response:** Thanks for your suggestion. We have described it in the caption. (Page 13, Line 272-273).

*Comments from Reviewer 3:*

*Comment 1:* *Please re-write and re-arrange sections 1 and 2: both sections introduced and reviewed the previous studies of nowcasting systems. The title of section 2 is "related work", which made me think it is a section of "methodology", but it seems that authors only did*

*literature review in section 2.1 for traditional echo extrapolation. Besides, although these traditional methods do not utilize amounts of historical images, it is capable of performing the nowcasting at least up to 1-hour. Some nowcasting systems combined different information and further improved the ability of nowcasting. I suggest authors to include those works in the literature review.*

*References:*

*1. Pulkkinen, S., Chandrasekar, V. and Niemi, T. (2021) Lagrangian integro-difference equation model for precipitation nowcasting. Journal of Atmospheric and Oceanic Technology, 38, 2125– 2145.*

*2. Chung, K.-S., and I. Yao, 2020: Improving radar echo Lagrangian extrapolation nowcasting by blending numerical model wind information: Statistical performance of 16 typhoon cases. Mon. Wea. Rev., 148, 1099–1120,*

*3. Nerini, D., Foresti, L., Leuenberger, D., Robert, S. and Germann, U. (2019) A reduced-space ensemble Kalman filter approach for flow-dependent integration of radar extrapolation nowcasts and NWP precipitation ensembles. Monthly Weather Review, 147, 987– 1006.*

**Response:** Thanks for your suggestion. We have rewritten Sections 1 and Section 2 based on the section formatting of the references you provided. The literature review is now all in Sections 1. We have also added the references you provided (Section 1; Page 2, Line 40-46).

**Comment 2:** *Lines 39-40, please clarify the meaning of "generating high-quality echo images".*

**Response:** Thanks for pointing it. High quality represents that the image is more realistic and structurally similar to the real image. We have clarified the meaning in the manuscript (Page 3, Line 68-69).

**Comment 3:** *Since this is a independent article, I suggest to briefly introduce the GAN-argcPredNet v1.0 in section 3.*

**Response:** Thanks for your advice. We have added the introduction of GAN-argcPredNet v1.0 (Section 2.1).

**Comment 4:** *Overall, the captions in the figures are not formal (by using "This is" ). In addition, the information is not clear enough in the captions.*

**Response:** Thanks for your suggestion. We have revised the description of the captions and added relevant information (Page 4, Figure 1; Page 6, Figure 2; Page 7, Figure 3; Page 8, Figure 4 and 5; Page 9, Figure 6; Page 10, Figure 7; Page 11, Table 1; Page 13, Figure 8; Page 14, Figure 9; Page 15, Figure 10; Page 16, Figure 11; Page 16, Table 2; Page 17, Table 3; Page 18, Figure 12; Page 19, Figure 13).

***Comment 5:*** *Authors introduced the quality control of radar data used in this study. Since it included 11 weather radar, I am curious that do authors encounter a problem of random interference in any weather radar? How to deal with this problem.*

**Response:** Thanks for your question. Our data have been professionally processed. We also verified the quality of the data before the experiment. You can find the descriptions in Page 10, Line 209-213.

***Comment 6:*** *Line 208: ".... with the height of 1km" please confirm that the "CAPPI" radar is used in this study.*

**Response:** Thanks for pointing it. We have added the description of "CAPPI" (Page 10, Line 218).

***Comment 7:*** *Z-R relationship has lots of uncertainties, do the coefficients of a and b obtain and determine in Guangzhou province? In addition, what kind of ground truth rainfall is used in this study? It is necessary to introduce it.*

**Response:** Thanks for your question. Our experiment is in cooperation with Guangdong Meteorological Observatory, and they have confirmed the experiment details. The kind of ground truth rainfall has been introduced in the manuscript (Page 6, Line 136).

***Comment 8:*** *Current study only compared the new systems to other deep-learning nowcasting systems. How is the performance compared to traditional radar echo extrapolations as mentioned in section 2.1?*

**Response:** Thanks for your advice. We have added a comparison experiment with the optical flow method (Page 12, Line 249-250; Page 12, Line 260-262; Page 13, Figure 8; Page 13, Line 276, Line 281, Line 284; Page 14, Line 286; Page 14, Figure 9; Page 15, Figure 10; Page 16, Figure 11; Page 16, Line 303; Page 18, Line 329-332).

***Comment 9:*** *The new nowcasting system is able to alleviate the echo attenuation compared to other deep-learning nowcasting models. However, can you say that GAN-argcPredNet v2.0 is able to capture the evolution (growth and decay) of the weather systems? The cases demonstrated in current study all focused on the growth of weather system, how about the decay of the weather system?*

**Response:** Thanks for your question. Based on the comments from the Topic Editor and Reviewer #1, we have revised the description of "intensify" (Page 1, Line 18, Line 20; Page 3, Line 91-92; Page 4, Line 111; Page 5, Line 119-120; Page 18, Line 341-343; Page 19, Line 363-366). We have also added an example of storm decay (Page 16, Figure 11). Our model still outperforms others.

***Comment 10:*** *Fig. 10 try to explain the reason of false predictions in Fig. 8. If you change the frequency or number of images as inputs, is it able to solve the issue?*

**Response:** Thanks for your question. The evolution process of rainfall is complex and variable, influenced by multiple factors. However, the rainfall evolution process reflected in the samples is limited. Our model performs better by maintaining the intensity, but due to the limitations of the samples, it cannot fully learn all the evolving features. As a result, deviations between the predicted trends and the real trends are inevitable, leading to false predictions. Even with an increase in the number of input images, it is still difficult to avoid false predictions.

.

---

## Author Response (AR3)

**RESPONSE LETTER**

We would like to thank the Topic editor and reviewers for the valuable suggestions on our manuscript once again. The suggestions made by the Topic editor and reviewers further improved our manuscript.

*Comments from Topic editor:*

**Comment 1:** *The authors have carefully revised their manuscript following the reviewers' comments and suggestions. However, as the reviewer suggested, please discuss the performance of very short-term nowcasting (0-30min) of GAN-argcPredNet v2.0 among all schemes.*

**Response:** Thanks for your suggestion. In comparison experiments, we add the average 0-30 min forecast lead-time scores and analyze the performance of each model from 0-30 min, 30-60 min forecast lead-time. The results show that GAN-argcPredNet v2.0 most effectively curbs echo attenuation and has a more competitive overall performance. (Page 12-13, Figure 8-9; Page 11-12, Line 258-275; Page 13, Line 284-298).

**Comment 2:** *Page 19, Line 351: This is because in the input sequence, the other strong echoes have a smaller range, making them more susceptible to attenuation during extrapolation. -> Did the authors mean the strong echoes have a smaller scale? Or, a smaller intensity range? Please clarify it.*

**Response:** Thanks for your suggestion. We modify it to "This is because in the input sequence, the other strong echoes have a smaller spatial range, making them more susceptible to attenuation during extrapolation." (Page 19, Line 361-362).

**Comment 3:** *Does this imply that the scheme learns only the main features and is less capable of tracking the minor features with rapid growth? If yes, please address this issue in the summary.*

**Response:** Thanks for your suggestion. We add the following description: "In Figs. 10-11, GAN-argcPredNet v2.0 has a clear advantage for out-of-center region curb the attenuation compared to other deep learning models. And there are fewer incorrect predictions compared to the optical flow. This indicates that GAN-argcPredNet v2.0 focuses on the main features and also pays more attention to the minor features with rapid growth than other models. Overall, GAN-argcPredNet v2.0 is capable of curbing echo attenuation, the center region is more significant, and the overall performance is more competitive." (Page 19, Line 362-366).

*Comments from Reviewer 1:*

*Comment 1: Page 1, line 17, "Aim to the problem,": This phrase sounds somewhat strange to me. Please consider rephrasing.*

**Response:** Thanks for your suggestion. We modify to "To solve this issue" (Page 1, Line 17).

*Comment 2: Page 3, line 85 and page 18, line 338: "more severe" -> "severer".*

**Response:** Thanks for your suggestion. We modify to "severer" (Page 3, Line 85; Page 18, Line 348).

*Comment 3:* The description of STIC is repeated too many times. I would recommend reducing them because this is redundant.

Page 3, line 91-92, "The purpose of the generator is to curb echo attenuation by suppressing the blurring effect of rain distribution and reducing the negative bias."

Page 4, line 111-112, "The STIC-Prediction generator is designed to reduce echo attenuation by suppressing the blurring effect and reducing the negative bias."

Page 5, line 118-119, "The purpose is to better maintain spatiotemporal features during information transmission within the model. After passing through STIC Attention, the features are fed to the lower layers, aiming to avoid blurring or to maintain the intensity during extrapolation."

Page 7, line 147-148, " The module calculates the importance of evolutionary information, which aims to suppress the blurring effect and reduce the negative bias."

In my opinion, the description in page 5 is sufficient. Please be concise.

**Response:** Thanks for your suggestion. We revise this description:

"The purpose of the generator is to more accurately forecast future precipitation distributions by curbing echo attenuation." (Page 3, Line 91-92).

"The STIC-Prediction generator is designed to reduce echo attenuation and consists of the argcPredNet and the STIC Attention module (Fig. 3)." (Page 4, Line 110-111).

"The module calculates the importance of evolutionary information." (Page 6, Line 147).

*Comment 4:* Figure 3, caption: The description here is somewhat confusing. Please consider rephrasing.

**Response:** Thanks for your suggestion. We have revised this description "The STIC-Prediction structure at time T with four layers (l=0, 1, 2, 3). The STIC Attention is located between the second layer (l=1) and the third layer (l=2). During prediction, $R_{l+1}^T$ forms the sequence

$R_{l+1}^0:R_{l+1}^T$ with $R_{l+1}$ prior to the moment T. Then, the sequence fed into the STIC module, which captures the correlation between the sequences and adjusts $R_{l+1}^T$. Finally, the new $R_{l+1}^T$ is output. See Section 2.2 for further explanation." (Page 6, Line 138-141).

**Comment 5:** *Page 12, line 250: "Pysteps" -> "pySTEPS"*

**Response:** Thanks for your suggestion. We revise to "pySTEPS"(Page 11, Line 248).

**Comment 6:** *In pySTEPS, what algorithm did you actually use? The pySTEPS has different options for motion vector computation. There must be also parameters for them.*

**Response:** Thanks for your suggestion. We add information about optical flow: "The first one is a traditional method, and the code comes from the pySTEPS library (Pulkkinen et al., 2019), which we performed using a local tracking approach (Lucas-Kanade)." (Page 11, Line 248-249).

**Comment 7:** *Regarding conventional spatiotemporal extrapolation methods, prediction accuracy can be largely improved by adopting data assimilation (you can find some previous studies). Although you do not need to perform additional experiments, it would be better to mention that in the discussion section. A fair comparison is always difficult; all the participating models must be well-tuned.*

**Response:** Thanks for your suggestion. We add an insight into data assimilation in the discussion section. "Given the proven efficacy of data assimilation in numerous fields, exploring the integration of data assimilation techniques with other meteorological variables, such as temperature, to study multi-modal models represents a crucial direction for precipitation nowcasting." (Page 20, Line 387-389).

*Comments from Reviewer 2:*

*Comment 1: About Fig.1 and Fig.2, since they both present the structure of GAN-argcPredNet v1.0 and v2.0 , it will be nice for the readers if authors arrange the same boxes、arrows、items in the same directions and positions. In that way, readers will be easier to recognize the differences between v1.0 and v2.0.*

**Response:** Thanks for your suggestion. We redrew Figures 1 and 2 (Page 4, Figure 1; Page 5, Figure 2).

*Comment 2: lines 281-285: please put the descriptions of Fig. 10 and Fig. 11 in the same paragraph as Fig. 9.*

**Response:** Thanks for your suggestion. We put together the descriptions of Figures 9,10,11. (Page 13, Line 284-298).

*Comment 3:* About the anlaysis of the results: so far, authors showed Figs. 9-11 as different cases to demonstrate different evolution of the events, and display the average scores in Fig. 8 (all events and 0-60 forecast leadtime) to illustrate that GAN-argcPredNet v2.0 outperform all other systems. However, by examining the performance of 0-30 min forecast lead-time (Figs. 9,10,11), it is also noticed that GAN-argcPredNet v2.0 may not always outperform other systems. For disaster prevention, it is very important to examine the performance of nowcast in a short period of time, and Fig. 8 somehow only reflects 1-h accumulated results. I think it is necessary to show and explain (or discuss) the results/performance between 0-30 min and 30-60 min forecast lead-time of the nowcasting systems. It also corresponds to the future work in the conclusion.

**Response:** Thanks for your suggestion. In comparison experiments, we add the average 0-30 min forecast lead-time scores and analyze the performance of each model from 0-30 min, and 30-60 min forecast lead-time. The results show that GAN-argcPredNet v2.0 most effectively curbs echo attenuation and has a more competitive overall performance. (Page 12-13, Figure 8-9; Page 11-12, Line 258-275; Page 13, Line 284-298).

---

## Author Response (AR4)

**RESPONSE LETTER**

We would like to thank the Topic editor and reviewers for the valuable suggestions on our manuscript once again. The suggestions made by the Topic editor and reviewers further improved our manuscript.

*Comments from Reviewer 3:*

*Comment: Figs. 8 and 9 show the skill score only at 30-min and 60-min nowcast lead-time based on my previous comments. However, it would be better if authors can evaluate and validate the performance of forecast in consecutive time. For instance, examine the performance of Figs. 10、11 and 12 quantitatively by all forecast skill-score, i.e. show the scores at 6-, 12-, 18-, 24-....54-, 60-min. By doing that, the advantage of different nowcasting system will be easier to illustrate recognize.*

**Response:** Thanks for your suggestion. In the comparison experiments, we included continuous skill score curves for three prediction examples and analyzed the performance of the model over a 0-60 minutes prediction lead time. The results show that GAN-argcPredNet v2.0 has a more competitive overall performance. (Page 17, Figure 13; Page 13-14, Line 294-301).